# Chitin Synthesis in Yeast: A Matter of Trafficking

**DOI:** 10.3390/ijms232012251

**Published:** 2022-10-14

**Authors:** Noelia Sánchez, César Roncero

**Affiliations:** Departamento de Microbiología y Genética, Instituto de Biología Funcional y Genómica, CSIC, Universidad de Salamanca, C/Zacarías González, s/n, 37007 Salamanca, Spain

**Keywords:** yeasts, chitin synthase, intracellular traffic

## Abstract

Chitin synthesis has attracted scientific interest for decades as an essential part of fungal biology and for its potential as a target for antifungal therapies. While this interest remains, three decades ago, pioneering molecular studies on chitin synthesis regulation identified the major chitin synthase in yeast, Chs3, as an authentic paradigm in the field of the intracellular trafficking of integral membrane proteins. Over the years, researchers have shown how the intracellular trafficking of Chs3 recapitulates all the steps in the intracellular trafficking of integral membrane proteins, from their synthesis in the endoplasmic reticulum to their degradation in the vacuole. This trafficking includes specific mechanisms for sorting in the *trans*-Golgi network, regulated endocytosis, and endosomal recycling at different levels. This review summarizes the work carried out on chitin synthesis regulation, mostly focusing on Chs3 as a molecular model to study the mechanisms involved in the control of the intracellular trafficking of proteins.

## 1. Chitin Synthesis: A Historical Perspective

Chitin is an essential polymer of the fungal cell wall which, due to its essentiality, has attracted considerable interest as a potential target for antifungal therapies. Therefore, it is not surprising that the mechanisms underlying chitin synthesis regulation have been under continuous scientific scrutiny.

Early on, researchers demonstrated that chitin synthesis in yeast occurs in a defined spatial and temporal sequence during the cell cycle [1]. Therefore, researchers have attempted to identify the molecular mechanisms that regulate chitin synthesis by biochemically analyzing the activity of yeast chitin synthases. Chitin synthases from different fungal origins behave as zymogens in vitro [2,3], and researchers have proposed multiple biochemical mechanisms for their activation, including allosteric regulation and activating and inhibitory factors. The most sustained hypothesis is based on the specific action of a dedicated protease, which was tentatively identified in yeast as proteinase B [4] and which is associated with an ill-defined inhibitory factor [5]. This model had persisted for almost a decade until the construction of the first proteinase B mutant [6], which is not affected in chitin synthase activity; hence, the role of proteinase B as an activating factor has been fully discarded.

Moreover, the mechanism regulating chitin synthase has to be fully compatible with the localization of the functional chitin synthase, a subject fiercely disputed over the years between the Cabib and Bartnicki-Garcia labs. Their views support two apparently incompatible localizations for active chitin synthases, namely the plasma membrane (PM) [7,8] and an intracellular compartment named chitosomes [9,10], respectively. The first insight into this dispute came from Randy Schekman, who envisioned an alternative explanation by placing chitosomes as secretory particles that transported zymogenic chitin synthase to the PM, where it would be activated by a then-unknown mechanism [11]. This hypothesis was later confirmed by the same group almost two decades later [12].

The arrival of molecular biology has allowed the identification of three different genes encoding chitin synthases in yeast. This information has promoted the generation of multiple mutants that have allowed analyzing their functions in yeast cells, providing several surprises (reviewed in [13]). The *CHS1* gene, the first to be identified, encodes the main chitin synthase activity in vitro (CSI) [14]; which has limited in vivo activity as a repair enzyme [15]. *CHS2* encodes a chitin synthase (CSII) with minor in vitro and in vivo activity: it is responsible for only 5–10% of total chitin synthesis [16]. However, this enzyme is essential in yeast biology: it is responsible for forming the chitin disk that separates mother and daughter cells after cell division [17]. The third chitin synthase gene, *CHS3*, was isolated several years later from mutants devoid of chitin [18]. It encodes a rather divergent chitin synthase (CSIII) responsible for 90% of the cellular chitin synthesis [19], although its function is not essential in yeast biology [17]. This functional scenario is maintained in most yeast, where proteins homologous to Chs2 and Chs3 have been characterized, while the situation in filamentous fungi is somehow different: they contain a larger number of *CHS* genes [20]. The secondary structure of the different types of chitin synthases identified to date is summarized in Figure 1A.

The main conclusion from the biochemical characterization of the major in vivo chitin synthase, Chs3, is that it is not truly zymogenic [21,22], a fact that calls into question the previous models regarding chitin synthesis regulation. Moreover, neither transcriptional nor translational regulation can account for Chs3 regulation, leaving posttranslational regulation as the functional alternative for chitin regulation [23]. The progressive characterization of multiple mutants that lack chitin and have allowed the identification of numerous proteins that are involved in the intracellular trafficking of Chs3 (reviewed in [13]) and have highlighted this protein as a paradigm in the study of the intracellular transport of proteins in yeast, which is the main subject of this review. Here, we approach the intracellular trafficking of Chs3 from its synthesis to its final delivery to the vacuole, highlighting the most distinctive steps of this process.

## 2. The Road of Chs3 to the PM and Beyond

Chs3 is a complex protein with many potential transmembrane (TM) domains (Figure 1B). Multiple studies with this protein indicate that it contains three cytosolic loops and a luminal region that is *N*-glycosylated, a structural model that has been confirmed by the AlphaFold model [24] (Figure 1C). This model predicts highly exposed regions on the N- and C-terminal domains of the protein that should facilitate the interaction of Chs3 with other proteins/complexes (see next chapters for further details on protein–protein interactions). The secondary structure of Chs3 resembles those of other complex polytopic proteins such as cellulose synthase [25] or CFTR [26]. The catalytic domain of the protein is at its central cytosolic region, with the substrate binding site just before a region containing several TM domains; this catalytic domain has recently been shown in CaChs2 to form a pocket involved in the creation of a membrane pore for chitin extrusion as it is synthesized [27]. The C-terminal cytosolic domain contains a conserved CHS domain (D-SWG) proposed to exert a regulatory role on chitin synthase activity [27], explaining the early report of the essentiality of this region for chitin synthase activity [28]. Chs3 is at a minimum *N*-glycosylated, ubiquitinated, palmitoylated, and phosphorylated (Figure 1B). These posttranslational modifications are necessary to establish a model of chitin synthesis regulation.

Despite its complex structure, Chs3 is precisely localized at the neck region of the cell during the cell cycle (Figure 2, wt images). Chs3 tagged with the green fluorescent protein (Chs3-GFP) appears at the bud neck in small-budded cells after bud emergence (G1–S phase), disappearing in medium-budded cells (G2 phase), to reappear in the bud neck of large-budded cells (M-phase), where it persists until the end of cytokinesis [29,30]. This timely localization is coincident with the onset of chitin synthesis [1]. However, some Chs3 remains in intracellular vesicles (chitosomes) during the cell cycle. In this review, we summarize the different steps of Chs3 intracellular trafficking (Figure 2), which is necessary for the enzyme to reach its functional localization, and discuss the molecular mechanisms that pave the road for Chs3 transport from the endoplasmic reticulum (ER) to the PM and beyond.

## 3. The First Step of the Road: Chs3 Folding and Export from the ER

Chitin synthases, including Chs3, lack a signal peptide sequence and its N-terminal region is cytosolic. Hence, the enzymes are type II TM proteins that are synthesized in the ER as a first step in their trafficking to the PM. We do not yet know any details about how Chs3 is synthesized by the ER-ribosomes or how cells achieve the folding of such a complex protein. However, Chs3 folding in the ER seems to be critical for its trafficking, because limited deletions in most regions of the protein lead to protein retention in the ER and a loss of protein functionality (Sanchez and Roncero, unpublished). Despite our limited knowledge of Chs3 folding, we now know how this protein is post-translationally modified in the ER and some of the rules that govern Chs3 exit from the ER.

Chs7 was identified from a mutant resistant to calcofluor and acts as a specific chaperone for Chs3 in the ER [31]. Chs7 exerts its function in a similar way to other dedicated chaperones such as Shr3, Pho86, and Gsf2 [32]. The interaction of these chaperones with their substrates prevents the aggregation of these proteins in the ER, allowing their export in COPII vesicles assisted by the Erv14 cargo receptor [33,34]. In the absence of the corresponding chaperones, these substrates are retained in the ER [32] (Figure 2, *chs7*∆ image), and most are subjected to strong ER-associated degradation (ERAD) [35]. However, Chs3 is not an apparent substrate for ERAD [31], a fact that initially explained its relatively long life [12]. Interestingly, Chs3 is *N*-glycosylated at several positions on its luminal domain, although the protein seems fully functional in the absence of glycosylation [34] (Sanchez and Roncero, unpublished). Our current work indicates that the non-glycosylated Chs3 protein is efficiently degraded by ERAD when retained in the ER (Sanchez and Roncero, unpublished); therefore, *N*-glycosylation shields the protein from ERAD recognition. This luminal domain, which is not present in the other chaperone-assisted proteins cited above, is essential for Chs3 trafficking because partial deletions of this domain retain protein in the ER, independently of its glycosylation status. Apparently, the potential sensitivity to ERAD provided by this essential luminal domain has been evolutionarily preserved though its *N*-glycosylation. Interestingly, the CFTR protein, with high structural similarity to Chs3, is also glycosylated on its luminal region, where it appears to have a relevant role on the ER quality control (ERQC) of the protein [26], potentially affecting its functionality.

Chs3 also oligomerizes in the ER through its N-terminal cytosolic domain [34]; this oligomerization is apparently maintained along the secretory route [36]. It is not yet known whether the Chs7 chaperone plays a role in Chs3 oligomerization, although some degree of oligomerization can be detected in the absence of the chaperone. This oligomerization is not essential for Chs3 trafficking or activity, but somehow it is required for the correct assembly of the chitin layer on the cell wall. Therefore, it is not surprising that Chs3 oligomerization is monitored in the Golgi apparatus [34]. Interestingly, protein oligomerization has also been described for cellulose synthase proteins to achieve proper cellulose synthesis within the plant cell walls (reviewed in [37]). Surprisingly, we have recently observed that the non-oligomerized protein can exit the ER independently of Chs7 or the Erv14 cargo receptor by a bulk flow mechanism (Sanchez and Roncero, unpublished results), suggesting that Chs7 directly participates in loading the functional Chs3 complexes into COPII vesicles. This interpretation is consistent with the formation of a molecular complex between Chs3 and Chs7, which has been recently described [38]. The Chs3-Chs7 complex is maintained along the secretory route; thus, it is very likely that the formation of this complex facilitates other steps involved in trafficking the functional protein to the PM (see next sections). Chs7 could also be required for Chs3 activity at the PM through an unknown mechanism [38]. The role of Chs7 as an assistant in Chs3 folding and function seems rather general because it has also been demonstrated in *Candida albicans* [39] and *Neurospora crassa* [40], but it is not apparently required for the folding and transport of other types of chitin synthases [20].

Chs3 is also palmitoylated [41] at its C-terminal region [42], although this palmitoylation does not seem fully necessary for Chs3 trafficking. The precise function of Chs3 palmitoylation on its intracellular trafficking remains unknown.

It is noteworthy that both the non-oligomerized and non-palmitoylated forms of Chs3 exit the ER efficiently, but they are recognized at the Golgi by the COPI machinery and returned to the ER [34,42]. This phenomenon indicates that the folding status of Chs3 is monitored in the Golgi in order to prevent the trafficking of incorrectly folded molecules. There are probably several mechanisms to detect misfolded Chs3 in the Golgi because there are different rules governing the retrieval of non-oligomerized and non-palmitoylated proteins. Non-oligomerized proteins are retrieved from the Golgi with the help of Rer1 [34] and Erv46 receptors (Sanchez, unpublished results), but no experimental evidence was found regarding the involvement of these receptors in the recycling of non-palmitoylated proteins [42]. Therefore, additional work with the different versions of Chs3 could help understand the mechanisms involved in retrograde trafficking from the Golgi through COPI vesicles.

In conclusion, Chs3 folding in the ER seems rather complex, as would be expected from the complex polytopic nature of the protein. What is rather unusual is the combination of multiple mechanisms that control its folding, mechanisms that until now have been independently characterized using different types of proteins. Therefore, Chs3 is emerging as an attractive model to decipher complex protein folding in the ER.

## 4. The *trans*-Golgi Network (TGN): The Central Station of Chs3 Trafficking

### 4.1. Going Ahead: Sorting Chs3 in the Golgi

Upon Chs3 arrival to the Golgi, the protein could be post-translationally modified in order to be delivered to its final destination. However, we know very little about these potential modifications in the Golgi during the anterograde transport of the protein. There are no doubts that the TGN is the central station in the sorting of Chs3 to its final destination. Chs3 exerts its function at the PM, where it is delivered from the Golgi in a temporally and spatially regulated manner. This delivery depends on the exomer, a TGN complex originally identified in yeast through the characterization of *chs5*∆ and *chs6*∆ mutants deficient in chitin synthesis [43,44,45]. Exomer facilitates the sorting of Chs3 in the TGN to promote its delivery to the PM using the general secretory mechanisms for polarized secretion, including Sec4 and most of the exocyst complex components [29].

The role of Chs5 and Chs6 in trafficking Chs3 from the Golgi was described in the late 1990s [46,47] (Figure 2, see *chs5*∆ image). However, exomer was described nearly a decade later in 2006: the Schekman and Spang labs independently identified this multiprotein complex [44,45]. These researchers have described how the GTP binding protein Arf1 recruits Chs5 to the TGN membrane to form the exomer complex together with four different Chs5p-Arf1p-binding proteins (ChAPs), Chs6, Bch2, Bud7, and Bch1, which form an evolutionarily related family of proteins conserved across fungi [48]. The Chs3 protein interacts specifically with the exomer through its ChAP component Chs6 [44,49], allowing the sorting of the cargo to the PM. Preliminary evidence also suggests that the maintenance of the Chs3-Chs7 complex is necessary for the recruitment by the exomer and export from the TGN (Sanchez and Roncero, unpublished). At that time, the structural nature of the complex was uncertain—exomer was even described as a novel coat to capture membrane proteins in the TGN en route to the cell surface [45]. However, the number of cargo molecules for this complex has remained surprisingly small over the years: only Chs3, Fus1 [50], and Pin2 [51] have been described as dependent on exomer for their arrival to the PM, questioning a general role of exomer as a coat complex.

The structure of exomer was finally solved by the Fromme lab based on crystallization. They have shown that exomer assembles as a tetrameric complex formed by two copies each of Chs5 and different ChAPs subunits [52,53]. In their subsequent work, they have discriminated between the role of different ChAPs, demonstrating a membrane-remodeling capacity for the Bch1 and Bud7 subunits, but not for Chs6 and Bch2, which are implicated in cargo sorting [54]. This function in cargo sorting has been extensively demonstrated for Chs6, which is a cargo adaptor specific to chitin synthesis through its specific interaction with Chs3 [49]. Moreover, the different ChAPs establish mutual relationships in the assembly of exomer, in which the ChAPs with membrane-remodeling capacities have stronger stabilization capabilities [55]. Interestingly, the different functional roles of exomer components were anticipated by a synthetic genetic array (SGA) analysis that highlighted the much more import role of Chs5 than each individual ChAP [56]. The multifactorial assembly of the different ChAPs results in a fine-tuned sorting complex that could efficiently respond to environmental changes (e.g., stress).

An important question remained unaddressed, namely the evolutionary conservation of such a complex system [44] for such a small number of cargo molecules. However, some insight on this subject has been provided by the discovery of a different type of exomer cargo, Ena1, which does not strictly depend on exomer for PM delivery but only for polarized transport [57]. This discovery explains some additional phenotypes associated with the exomer mutants [44], providing a new scenario for the role of exomer in the TGN. Recently, the adaptor ubiquitin ligase Rcr1 has also been shown to behave as a non-canonical exomer cargo [58]. In this scenario, exomer cannot simply be visualized as an adaptor cargo complex; rather, it is a complex with additional functions in the TGN and is thus involved in the sorting and transport of multiple proteins. This wider role has been demonstrated in *Saccharomyces cerevisiae* [57,58,59] and other fungi [60,61], therefore explaining the evolutionary conservation of exomer components that can be traced to eukaryotic ancestors [62]. In most fungi, exomer is assembled only from the Chs5 and Bch1 subunits [48], performing its function in coordination with other TGN complexes such as AP-1 and GGAs, among others, participating in the polarized transport of several proteins. A gene duplication event originated Chs6, whose function is redundant with Bch1. Only the late whole genome duplication within the *Saccharomyces* genus allowed the specialization of some ChAPs as cargo adaptors for a dedicated set of proteins such as Chs3 or Pin2. Not surprisingly, the absence of exomer has only a moderate effect of chitin synthesis in other fungi [48]. Apparently, exomer is of general importance for fungal biology, however, the original model of exomer as a cargo adaptor is a distinct evolutionary characteristic of the *Saccharomyces* genus.

In conclusion, the work on chitin synthesis regulation in *S. cerevisiae* has allowed the identification of a new and unexpected complex involved in protein sorting in the TGN that plays a general role in eukaryotes. The current model proposes a general role for the exomer in protein polarization in fungi, whose importance differs depending on the cellular type, from strong in yeast cells to minor in hyphal cells [59], where the major role in polarization has been assumed at the TGN by the AP-1 complex [63], as well as the major complex involved in protein polarization in higher eukaryotes [64]. An interesting point on exomer function is that its requirement for chitin synthesis can be bypassed by eliminating other TGN sorting complexes including AP-1 [65] or GGA [66], a subject that is addressed in the next section.

### 4.2. Endosomal Recycling: A Noteworthy Step in the Intracellular Trafficking of Chs3

The potential role of endocytosis in the traffic of Chs3 was previously envisioned by Schekman’s group when they identified chitosomes, the intracellular accumulation of Chs3 in the Golgi, as endocytically derived particles [67]. How this intracellular reservoir originates is a crucial part of Chs3 biology.

Chitosomes are endocytically derived, but they also accumulate in the *chs5*∆ and *chs6*∆ exomer mutants [46,47]; therefore, they are derived from both anterograde and retrograde transport. The Schekman lab also provided an explanation for this phenomenon since they demonstrated that endocytosed Chs3 is recycled from the early endosome (EE) to the TGN by the AP-1 complex [65]. While this conclusion is still valid, the existence of the EE compartment in yeast has been questioned [68]; hence, this recycling probably occurs within the TGN boundaries. Nevertheless, recycling prevents the trafficking of Chs3 to the vacuole and explains the early results showing the long life of this protein [12]. In addition, recycling generates an extensive population of Chs3 in the TGN that merges the recycled protein with the de novo synthesized protein. The Chs3 recycling from the TGN/EE is mediated by the direct interaction between the AP-1 complex and the N-terminal cytosolic domain of Chs3 [69]. Of note, it also depends on epsins (Ent3/5) and GGA complexes [66], probably in an indirect way because of the heavily interconnected roles of these complexes in the TGN [70]. An interesting feature of this system is that in the absence of AP-1, Gga1/2, or Ent3/5, Chs3 can travel to the PM independently of exomer through a poorly described alternative route [65,66]. The rerouting of Chs3 in the absence of GGAs is similar to what has been described for many other proteins such as the amino acid permeases Gap1 and Tat2, but it is mechanistically different because these proteins do not directly interact with AP-1, and their trafficking is thus fully independent of this complex [59]. The current model suggests that exomer and AP-1 compete for Chs3, dictating the amount of Chs3 that is delivered to the PM, where the modification of its residence time via endocytosis (see next section) contributes to the level of chitin that is synthesized.

This model is conceptually very similar to the trafficking of many permeases, whose endosomal recycling is regulated based on nutrient availability (reviewed in [71]). In the case of chitin synthesis, this regulation was linked to cell cycle progression [30] and to increased temperature [72] and cell wall stress [73]. These conditions increase the Chs3 accumulation at the PM and promote the upregulation of chitin synthesis to allow the cell to survive cellular stress.

After the initial work from the Schekman Lab, it has become clear that the TGN is the principal station involved in Chs3 trafficking. Thus, it is not surprising that Chs3 has been widely used as a molecular tool to study the different aspects of endosomal trafficking, including the role of lipid synthesis/asymmetry [74,75,76], or the role of Rab GTPases [77,78] and Snare proteins [79,80]. In addition, Chs3 has been used to characterize alternative forms of the AP-1 complex in yeasts [81].

## 5. Controlling Residence Time Is the Chief Regulator of Chitin Synthesis

Chitin synthases synthesize chitin in a vectorial mode [82], getting the enzymatic substrate from the cytoplasm and extruding the nascent chitin to the periplasmic space. This model has been fully confirmed by the recent crystallization of an active CaChs2 protein [27]. This specific topology is not possible in chitosomes, where chitin synthases remain conformationally inactive, becoming active only after its insertion into the PM.

After Chs3 sorting in the TGN, the protein is delivered to the PM in a polarized form, localizing at the mother side of the neck region [46]. This localization is dependent on several cellular components including septins, Bni4, and Chs4 [83,84,85]. Chs3 interacts directly with Chs4, which physically interacts with Bni4, anchoring the Chs3/Chs4 complex to the septin ring [83]. Our group demonstrated that Chs4 is not required for Chs3 delivery to the PM [86], but rather it is required to prevent/delay the endocytosis of the protein through its anchor to the septin ring, which delays the arrival of Chs3 to the zone of active endocytosis defined by actin patches assembled around the neck [87]. Chs4 is prenylated [88]; prenylation is not required for Chs3/Chs4 interaction, but it favors Chs4 interaction with membranes and its action on Chs3 [86]. Preliminary evidence suggests that Chs4 prenylation could affect the lateral displacement of the active Chs3/Chs4 complex (Sacristan and Roncero, unpublished) in accordance with the general role of prenylation on the lateral displacement of proteins along membranes [89]. This would affect Chs3 endocytosis contributing to the localized synthesis of chitin.

In addition to its anchoring function, Chs4 has also been described as a posttranslational activator of Chs3 [22,90,91]. Interestingly, Chs4 does not function as an activator of chitin synthesis during sporulation because it is rapidly degraded and replaced by Shc1, a close homolog that is induced during spore formation [92]. The activating function of Chs4/Shc1 is essential for Chs3 functionality; therefore, during vegetative growth, Chs3 remains inactive in the absence of Chs4 despite its delivery to the PM, resulting in the absence of chitin synthesis. Although Chs3 and Chs4 colocalize in cells during most of the cell cycle [30,93], Chs3 and Chs4 can arrive to the PM independently of each other. Hence, we do not know the precise time for their interaction and thus the precise time for the induction of chitin synthase activity. Chs4 physically interacts with the middle cytosolic domain of Chs3 [36,86,94], and this interaction has been proposed to be modulated by Hof1 to regulate the chitin synthesis during cytokinesis [94]. However, the precise mode in which Chs4 activates the chitin synthase activity remains unknown. The recent crystallization of the first chitin synthase allows a tentative hypothesis for Chs4 function. The catalytic domain of Chs3 is preceded by a TM segment and this structure probably prevents the protein from adopting the active form, namely a completely globular catalytic domain (based on the recent description for CaChs2). The interaction between Chs4 and the catalytic cytosolic domain of Chs3 would allow the protein to acquire an active form similar to that described for CaChs2. A similar effect is produced by the treatment of membrane extracts with trypsin, which elicits a strong increase in chitin synthase activity in vitro in the *chs4*∆ mutant [90].

Considering all this evidence, the role of Chs4 in Chs3 trafficking can be summarized as follows: Chs4 interacts with Chs3, favoring its correct insertion into the PM in a conformationally active form. This interaction also facilitates the anchorage of the active complex to the septin ring, delaying the endocytosis of the protein and regulating the permanence of active Chs3 at the neck and the onset of chitin synthesis. A delay in the endocytic turnover of Chs3 serves to sustain the increased chitin synthesis required during hyperpolarized growth in yeast cells [87]. It remains to be tested whether the role of Hof1 regulating Chs3 activity [94] is simply achieved by modulating its endocytic turnover through Chs4. Moreover, a general reduction in cellular endocytosis by chemical or genetic methods leads to Chs3 accumulation at the PM (Figure 2, *end*∆ image) and significantly increases chitin synthesis [86,87,95]. Thus, the cellular control of Chs3 endocytosis allows yeast cells to control the cellular levels of chitin in order to adapt their growth to changing environmental conditions.

The function of Chs4 on class IV chitin synthases has apparently been conserved in fungi, because the *chs4*∆ mutants of *C. albicans* [96] and *Cryptococcus neoformans* [97] show similar phenotypes associated with the major absence of chitin in their cell walls. Unfortunately, the potential role of Chs4 orthologs [20] in the function of class V/VII chitin synthases has not yet been investigated.

## 6. Avoiding the Vacuole: A Signature of Chs3 Trafficking

A question that had been unaddressed for years was whether Chs3 is trafficked to the vacuole. Researchers appeared to ignore this potentiality because of the minor degradation of the protein and its effective retention in the TGN [65]. However, we have very recently shown that Chs3 is effectively trafficked to the vacuole for degradation [98]. The amount of Chs3 that follows this route is limited but not insignificant, and it is associated with the ubiquitination of Chs3 at the PM mediated by the Art4 arrestin and the Rps5 ubiquitin ligase. This ubiquitination occurs at the N-terminal cytosolic domain of the protein and has no apparent effect on protein endocytosis. Rather, it serves as signal recognition for the ESCRT complex, which directs protein from the endosomal compartment to vacuolar degradation (Figure 2, see *vps27*∆ image). How ubiquitinated protein escapes AP-1 retention in the TGN is unclear, but it should be considered that both ubiquitination and AP-1 binding domains are located at the same N-terminal region of the protein (Figure 1B), probably causing interference. This trafficking to the vacuole is fully consistent with Chs3 homeostasis through the degradation of the damaged protein that was exposed to the extracellular media, as has been demonstrated for other proteins [99].

Surprisingly, the non-ubiquitinated version, Chs3 also reaches the vacuole, but it is not efficiently internalized to its lumen, accumulating at the vacuolar membrane. This accumulation increases in the retromer mutant *vps35*∆ [98] (Figure 2, *vps35*∆ image), indicating that the retromer complex is able to recycle Chs3 from the late endosomal compartment to the TGN, as later demonstrated in an independent study [100]. This complex directly interacts with Chs3 through its N-terminal cytosolic region, probably through several independent regions [98,100]. This interaction is apparently independent of ubiquitination because the retromer is able to efficiently recycle both the ubiquitinated and the non-ubiquitinated proteins. The total amount of Chs3 that travels through the late endosomal compartment to the vacuole is unknown; however, the mild phenotypes associated with ESCRT and retromer mutants in terms of chitin synthesis [98] suggest that trafficking through this route is minor, contributing more to Chs3 homeostasis than to the effective regulation of chitin synthesis.

More recently, an alternative mechanism for Chs3 degradation in the vacuole after Ca^2+^ stress has been demonstrated. In this case, the induction of ubiquitin ligase adaptor Rcr1 promotes Chs3 ubiquitination at the PM, leading to its rapid degradation in the vacuole [58]. The nature of this ubiquitination or the specific mechanisms for Chs3 trafficking to the vacuole remain unexplored.

In conclusion, the synthesis of chitin along the cell cycle of *Saccharomyces* yeast is regulated from an intracellular reservoir of Chs3, the chitosomes, which is maintained not only through continuous de novo synthesis of Chs3, but also by its endocytic turnover. This turnover is determined by the endocytosis rate and the endocytic recycling of the protein, effectively performed from different endosomal compartments by the AP-1 and retromer complexes. In this scheme, protein degradation seems to act as a minor default mechanism, as interpreted in an earlier report [65].

## 7. Chs3 Outside of Yeast Cells

A newly emerging subject in the study of Chs3 transport is its presence in extracellular vesicles (EVs) outside yeast cells [101]. These Chs3-containing vesicles were demonstrated to improve in trans the growth of cell wall-defective mutants [101], supporting the notion that Chs3 uptake via EVs could have an important role in cell wall remodeling. The origin of these EVs is still uncertain, but it has been linked to different steps of the secretory pathway (reviewed in [102]), opening new perspectives for further studies in Chs3 trafficking.

## 8. Other Chitin Synthases and Other Modes of Intracellular Transport

While we mostly discussed Chs3—because it is responsible for the majority of chitin synthase activity in yeast—these organisms also depend on Chs2 for the synthesis of the chitin disk, which forms the primary septum that is essential for cell division. In this case, the regulation of Chs2 activity is also dependent on intracellular transport, but it follows completely different rules.

Chs2 is synthesized in a cell cycle-regulated mode [23,103], but the coordination of its function with cell cycle progression is potentiated by a specific retention mechanism of the protein in the ER [104]. Chs2 is phosphorylated by the CDK1 activity at its N-terminal cytosolic domain, preventing its export from the ER [104]. Only an increase in the activity of Cdc14 phosphatase at the end of mitosis promotes Chs2 dephosphorylation [105] and allows its incorporation in COPII vesicles [106], thus leading to its final delivery to the neck region by the general secretory mechanism (reviewed in [107]). The cell polarity protein Spa2 directs the incorporation of Chs2 into the so-called “ingression progression complex” formed also by Hof1, Inn1, and Cyk3 [108], which directly regulates the rate of primary septum in close coordination with the contraction of the AMR [109,110]. This complex also prevents the premature activation of Chs3 [94]. Finally, Chs2 is simply inactivated by endocytosis and further transport to the vacuole. The much simpler topological structure of Chs2 compared with Chs3 explains the much more direct regulation of its transport and the absence of dedicated proteins required for it. No recycling mechanism was described in yeast for Chs2, but the homologous ChsB of *Aspergillus nidulans* seems to be endocytically recycled to maintain its functional activity at the PM [111]. Therefore, the regulation of Chs2 trafficking by endocytic recycling should be re-evaluated in the future.

The function of Chs2 during cytokinesis was conserved during fungal evolution; hence, it is very likely that there are conserved mechanisms controlling its regulation. However, filamentous fungi contain additional different classes of chitin synthases that seem to travel in different types of intracellular vesicles [112]. Class V and VII chitin synthases are topologically similar to Chs3, but they contain a myosin motor-like domain (MMD) at their N-terminal regions that substitute the simple cytosolic N-terminal region of Chs3 (Figure 1A). They do not require dedicated ER chaperones or exomer for their trafficking because mutants of the conserved *CHS7* [113] or *CHS5/6* [40] genes do not affect their function. Thus, we should assume that the presence of the MMD strongly affects their trafficking compared with Chs3, simplifying the requirements for distinct, dedicated proteins. These enzymes travel to the hyphal tip in intracellular vesicles through central microtubules [114]. This transport is independent of the MMD domain, which is later required for their final delivery to the PM through an actin-dependent mechanism. The MMD domain of the protein finally enhances polar secretion by tethering vesicles at the site of exocytosis [114]. Interestingly, MMD-containing chitin synthases also facilitate the delivery of other cell wall synthases traveling in the same vesicles [115]. In summary, intracellular trafficking of these enzymes is linked to chitin synthesis as part of the fungal cell wall.

## 9. Concluding Remarks

We described how chitin synthesis is regulated in fungi. This regulation is a direct consequence of the control of the intracellular trafficking of the proteins involved in its synthesis. Chs3 has become a paradigm for studies of the intracellular trafficking of proteins because its trafficking basically recapitulates all the previously described steps on the secretory route, from the ER to the vacuole. Thus, it represents an excellent model for these studies. This complex protein contains multiple signals that regulate its trafficking, therefore providing an integral model to study the interactions between the different mechanisms that control the intracellular trafficking of proteins. We expect that additional studies in the field could help to address some remaining question in field, such as the role of palmitoylation in the biology of integral TM proteins or the mechanisms involved in the recognition of misfolded proteins in the Golgi.

## Figures and Tables

**Figure 1 ijms-23-12251-f001:**
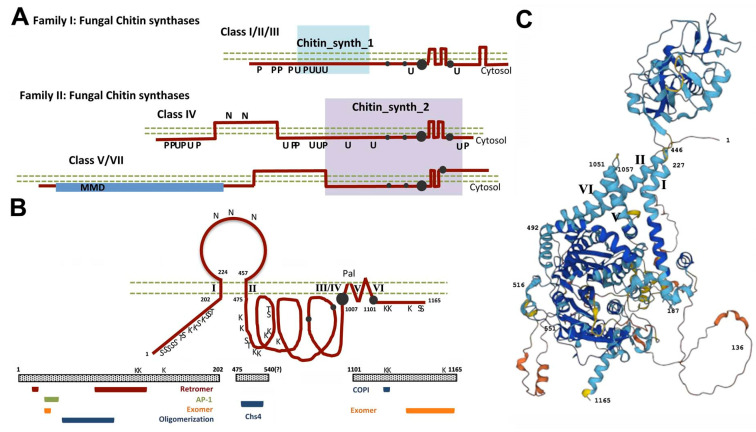
Structure of the chitin synthase (Chs) proteins. (**A**) Classification of the different Chs proteins based on their amino acid sequences and predicted secondary structures. Note that these proteins are expected to be type II integral transmembrane proteins. The different domains of each protein are indicated together with the conserved catalytic regions (gray dots). Posttranslational modifications are indicated based on the high-throughput analysis performed for the *Saccharomyces cerevisiae* proteome. Phosphorylation (P), ubiquitination (U), and *N*-glycosylation (N) are noted. (**B**) Detailed structure of the *S. cerevisiae* Chs3 protein with its three cytosolic domains and a luminal region that is *N*-glycosylated. The transmembrane domains are depicted with their predicted architecture, including the proposed site for palmitoylation (Pal). All the phosphorylation (S/T) and ubiquitination (K) sites that were experimentally verified are indicated. The lower part of the figure shows all the protein regions that were shown to be required for interaction with the indicated complexes or proteins. Note that most of them are located at the N-terminal cytosolic region of the protein, which serves as a major platform mediating protein interactions. (**C**)The predicted AlphaFold model structure for Chs3. This model confirms the proposed structure with a luminal region (upper part) and three cytosolic regions with some domains highly exposed and thus susceptible to performing protein–protein interactions. The four TM domains that cross the membrane are neatly visible and are labeled as in (**B**). See the text for more detailed descriptions of the roles of each domain.

**Figure 2 ijms-23-12251-f002:**
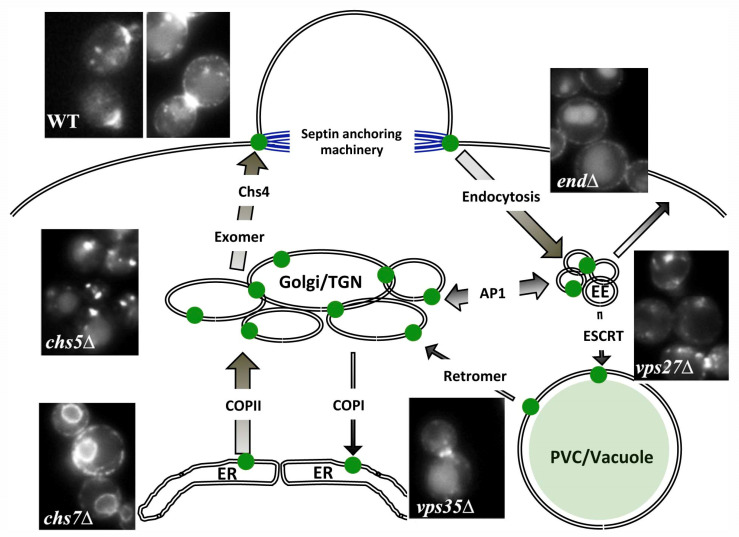
A visual scheme for chitin synthase 3 (Chs3) trafficking. Chs3 is synthesized in the endoplasmic reticulum (ER) where it is folded and later transported to Golgi in COPII vesicles. Misfolded proteins can be detected in the Golgi and returned to the ER via COPI vesicles. Chs3 is sorted in the *trans*-Golgi network (TGN) using exomer in order to be transported to the plasma membrane (PM) in a polarized way. On this journey, it interacts with Chs4, which elicits chitin synthase activity, allowing its anchorage to the neck region through specific machinery. Subsequently, the protein is endocyted, being recycled to the TGN from the early endosomal compartment by the AP-1 complex. In the absence of this recycling, Chs3 can escape to the PM though an alternative route. Only a small amount of the protein, which was ubiquitinated at the PM, is recognized by the ESCRT complex and delivered to the vacuole for degradation. Part of this protein can be recycled back to the TGN by retromer, avoiding degradation. Because of these efficient recycling mechanisms, most Chs3 accumulates in the TGN, forming a dedicated intracellular reservoir known as chitosomes. The early endosomal (EE) compartment has been depicted here for clarity, but its existence in yeast is questionable, and it is very likely that recycling by the AP-1 occurs within the TGN boundaries. The width of the arrows reflects the intensity of the traffic. The images show Chs3- GFP localization in the wild type (WT) and the indicated mutants: Chs3 accumulates normally at the neck region and in intracellular vesicles (chitosomes), while images from the mutants show the accumulation of the protein at the different intracellular compartments. *chs7*∆: ER; *chs5*∆: TGN/chitosomes; *end*∆: PM; *vps27*∆: endosome E; *vps35*∆: vacuole.

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
