# Peer review of "Chitin Synthesis in Yeast: A Matter of Trafficking"

_ijms, 2022, doi:10.3390/ijms232012251_

Round 1
Reviewer 1 Report
This manuscript from Sanchez and Roncero is a well-written and comprehensive review of the factors regulating the transport of the chitin synthase Chs3 to the cell surface in S. cerevisiae. Given the complexity of this process, the study of Chs3 transport provides insights into regulated transport more generally and should be of broad interest.
Major comment:
There is a nice, high confidence structural model of Chs3 available from Alphafold. It would be very valuable and informative to have a figure that maps the various interacting regions and the predicted catalytic region of Chs3 described in the text onto that model rather than the cartoon shown in Figure 1.
Minor comments and edits:
Line 9 “pioneering” not pioneer
Line 21 insert “to”: “…due to its…”
Line 50 insert “a”: “…as a repair…”
Line 66 information missing from between parentheses
Figure 1 – it’s not clear what region of Chs3 the blue bar labeled Chs4 is supposed to indicate.
Line 80 what is “transductional regulation”? should this be “translational”?
Figure 2: It would be easier to see the differences in distribution if the images of the cells were larger and showed and perhaps showed only a single cell.
Line 161: What is the citation for “high structural similarity” of CFTR to Chs3? This seems surprising given that CFTR is a choride channel and the predicted structures by Alphafold do not display significant similarity.
Line 182. Delete “general” (redundant with the next line).
Line 281 “It should be clear from the above...” I don’t think the importance of endocytosis is clear to the reader at this point. The word “endocytosis” hasn’t even been mentioned in the text yet. A better transition is needed.
Line 384: remove extra space “arrestin and”
Line 404 “suggests” should be “suggest”
Line 413 should be “an intracellular reservoir”
Author Response
We really appreciate the comment on the AlphaFold model for Chs3, although we are not sure about the utility of such a model in a printed version, which only show a single view of the model. Anyhow, we have included the AlphaFold model of Chs3 in the best orientation possible (Figure 1C). The model confirms the proposed model that is presented is the less refined, but still functional, cartoon of Figure 1B. The new figure 1C highlights the exposed cytosolic regions that serve as platforms for protein-protein interactions as they are later described along the manuscript (see new Figure 1).
Minor points:
Line 9 “pioneering” not pioneer
Done
Line 21 insert “to”: “…due to its…”
Done
Line 50 insert “a”: “…as a repair…”
Done
Line 66 information missing from between parentheses
Information is now included
Figure 1 – it’s not clear what region of Chs3 the blue bar labeled Chs4 is supposed to indicate.
Figure 1 has been modified accordingly.
Line 80 what is “transductional regulation”? should this be “translational”?
Yes, Done
Figure 2: It would be easier to see the differences in distribution if the images of the cells were larger and showed and perhaps showed only a single cell.
Images have been enlarge as much as possible. Most of them contain a single cell.
Line 161: What is the citation for “high structural similarity” of CFTR to Chs3? This seems surprising given that CFTR is a choride channel and the predicted structures by Alphafold do not display significant similarity.
We never attempted to compare the structures of such a divergent proteins, but only to highlights the presence of alternating cytosolic, lumenal and TM regions that necessarily have to be properly folded following similar rules. The text has been modified in order to diminish the structural relevance of the comparison.
Line 182. Delete “general” (redundant with the next line).
Done
Line 281 “It should be clear from the above...” I don’t think the importance of endocytosis is clear to the reader at this point. The word “endocytosis” hasn’t even been mentioned in the text yet. A better transition is needed.
We agree, the text has been modified
Line 384: remove extra space “arrestin and”
Done
Line 404 “suggests” should be “suggest”
Done
Line 413 should be “an intracellular reservoir”
Done
Reviewer 2 Report
This review of chitin synthesis regulation in fungi is focused on the yeast S. cerevisiae and on intracellular trafficking of the enzymes responsible for chitin synthesis. It is very thorough and well written, and will be of great value to the field. I have only some minor comments (below) to improve clarity.
Line 33: I recommend exchanging “sustained” for “survived” or “persisted”.
Line 34: “which is not affected by chitin synthase activity”: unless I am misunderstanding something, I think the correct wording should be: “which did not affect chitin synthase activity”.
Line 63: “classification of the different classes” sounds redundant. Perhaps change to “Classification of the different Chs proteins”.
Line 66: What are the parentheses intended to represent here?
Line 67: I’m unsure about the intended meaning of “massive” in this sentence. Perhaps “high-throughput” or “mass spectrometry”?
For the illustrations in Figure 1, I recommend removing the “shadow” effect for all objects. Some objects lack the shadow, such as the presumptive cytosolic domains of Chs3 in panel B, but it is not clear if this difference is intended to convey any meaning, hence the shadow effect is distracting. Also, what are the gray circles in the illustrations? Are these the sites of palmitoylation?
Line 91: I am unsure of what “less sustained” here is intended to mean. Is it that these models are less supported by experimental data? Or that these models were proposed at one point but are no longer seriously considered?
Line 94: delete “of” in “before of”
Line 103: I recommend adding “of the cell” after “neck region”, to make it clear that “neck” does not refer to the constriction of an endocytic vesicle, for example.
Line 127: Presumably “EE” refers to early endosome?
Similarly, Line 132: “endosome E”?
Line 138: instead of “ synthesized in the ER-ribosomes” I recommend “ synthesized by ER ribosomes”
Line 178: a personal preference: I suggest “Chs3–Chs7” as a way to indicate the complex, rather than “Chs3/Chs7”, which could be misunderstood as referring to two alternative names for Chs3.
Lines 182-183: The word “general” appears twice in this sentence; I suggest removing the first instance.
Lines 201-202: “as one would be expected”: change to either “as one would expect” or “as would be expected”.
Line 212: “Chs3 exerts its function on the PM”: I read this as saying that Chs3 modifies the PM in some way. I recommend changing to “Chs3 exerts its function at the PM”.
What are “GGAs”? Can these be defined somewhere?
Lines 412-413: This sentence is very long, with multiple “which” statements, and includes “regulation […] is regulated”, which is awkward wording. I recommend breaking up the sentence and rewording. Also, “a intracellular” should be “an intracellular”.
The title of section 7, “Chs3 Beyond Yeast Cells”, is a little misleading. I was expecting some discussion of Chs3 homologs in non-yeast fungi. Perhaps “outside of” would be a good replacement for “Beyond”?
Line 449: “revaluated” should be “re-evaluated”.
A naive question: is the myosin-motor-like domain (MMD) actually thought to somehow involve direct motility along actin cables?
Author Response
Line 33: I recommend exchanging “sustained” for “survived” or “persisted”.
Done
Line 34: “which is not affected by chitin synthase activity”: unless I am misunderstanding something, I think the correct wording should be: “which did not affect chitin synthase activity”.
Sorry, it was a mistake. The text has been modified.
Line 63: “classification of the different classes” sounds redundant. Perhaps change to “Classification of the different Chs proteins”.
Done
Line 66: What are the parentheses intended to represent here?
Figure legend has been modified.
Line 67: I’m unsure about the intended meaning of “massive” in this sentence. Perhaps “high-throughput” or “mass spectrometry”?
Agree, high-throughput is fine and the text has been modified.
For the illustrations in Figure 1, I recommend removing the “shadow” effect for all objects. Some objects lack the shadow, such as the presumptive cytosolic domains of Chs3 in panel B, but it is not clear if this difference is intended to convey any meaning, hence the shadow effect is distracting. Also, what are the gray circles in the illustrations? Are these the sites of palmitoylation?
Figure 1 has been extensively modified, including the figure legend in order to satisfy this and other requests.
Line 91: I am unsure of what “less sustained” here is intended to mean. Is it that these models are less supported by experimental data? Or that these models were proposed at one point but are no longer seriously considered?
During the assembly of the manuscript we revised alternative folding models for Chs3, which typically include additional TM and luminal domains. We really believed that these models were poorly sustained by experimental evidence, as it is also apparent from the AlphaFold model recently published, and now included in Figure 1C, that strongly support our structural model for Chs3 (Figure 1B). The new redaction of the text makes unnecessary the reference to other folding models.
Line 94: delete “of” in “before of”
Done
Line 103: I recommend adding “of the cell” after “neck region”, to make it clear that “neck” does not refer to the constriction of an endocytic vesicle, for example.
Done
Line 127: Presumably “EE” refers to early endosome?
Yes, as it is now indicated in the legend.
Similarly, Line 132: “endosome E”?
I do not understand this comment.
Line 138: instead of “ synthesized in the ER-ribosomes” I recommend “ synthesized by ER ribosomes”
Done
Line 178: a personal preference: I suggest “Chs3–Chs7” as a way to indicate the complex, rather than “Chs3/Chs7”, which could be misunderstood as referring to two alternative names for Chs3.
Done
Lines 182-183: The word “general” appears twice in this sentence; I suggest removing the first instance.
Agree, word has been removed.
Lines 201-202: “as one would be expected”: change to either “as one would expect” or “as would be expected”.
Corrected.
Line 212: “Chs3 exerts its function on the PM”: I read this as saying that Chs3 modifies the PM in some way. I recommend changing to “Chs3 exerts its function at the PM”.
Agree, corrected.
What are “GGAs”? Can these be defined somewhere?
Obviously, GGAs form part of a Golgi complex involved in endosomal traffic like AP-1, but It is not easy to introduce briefly GGAs and the description of the acronym (Golgi-localized, Gamma-adaptin ear homology, Arf-binding protein) is here of little help. Therefore we should maintain the text, being aware that are many other components of the secretory route that are also only cited. This manuscript is not a review of the secretory route, but rather a picture of the travel of Chs3 along it.
Lines 412-413: This sentence is very long, with multiple “which” statements, and includes “regulation […] is regulated”, which is awkward wording. I recommend breaking up the sentence and rewording. Also, “a intracellular” should be “an intracellular”.
The sentence has been modified.
The title of section 7, “Chs3 Beyond Yeast Cells”, is a little misleading. I was expecting some discussion of Chs3 homologs in non-yeast fungi. Perhaps “outside of” would be a good replacement for “Beyond”?
Agree, title has been modified.
Line 449: “revaluated” should be “re-evaluated”.
Done
A naive question: is the myosin-motor-like domain (MMD) actually thought to somehow involve direct motility along actin cables?
The question about the function of MMD domain of fungal CHSs is complicated. The domain is essential for PM delivery, but it is not clear its role on the transport along the cell. Most long-range vesicular traffic in fungi occurs through microtubules (MT) and definitively the MMD domain is not required for this traffic. However, MMD domain interacts with F-actin, probably facilitating the polar traffic of Chss in the sort range. As far I know, Gero Steinberg concluded that the MMD domain supports exocytosis but not long-range delivery of transport vesicles.